# A Peptide Inhibitor of Peroxiredoxin 6 Phospholipase A_2_ Activity Significantly Protects against Lung Injury in a Mouse Model of Ventilator Induced Lung Injury (VILI)

**DOI:** 10.3390/antiox10060925

**Published:** 2021-06-07

**Authors:** Aron B. Fisher, Chandra Dodia, Shampa Chatterjee

**Affiliations:** Department of Physiology and Institute for Environmental Medicine, University of Pennsylvania Perelman School of Medicine, Philadelphia, PA 19104, USA; cdodia@pennmedicine.upenn.edu (C.D.); shampac@pennmedicine.upenn.edu (S.C.)

**Keywords:** acute lung injury (ALI), lung oxidant stress, NADPH oxidase type2 (NOX2), reactive O_2_ species (ROS), peroxiredoxin 6 inhibitor peptide-2 (PIP-2)

## Abstract

Ventilator induced lung injury (VILI) is a lung injury syndrome associated with mechanical ventilation, most frequently for treatment of Acute Lung Injury (ALI), and generally secondary to the use of greater than physiologic tidal volumes. To reproduce this syndrome experimentally, C57Bl/6 mice were intubated and ventilated with low (4 mL/Kg body weight) or high (12 mL/Kg) tidal volume for 6 h. Lung parameters with low volume ventilation were unchanged from non-ventilated (control) mice. High tidal volume ventilation resulted in marked lung injury with increased neutrophils in the bronchoalveolar lavage fluid (BALf) indicating lung inflammation, increase in both protein in BALf and lung dry/wet weight indicating lung edema, increased lung thiobarbituric acid reactive substances (TBARS) and 8-isoprostanes indicating lung lipid peroxidation, and increased lung protein carbonyls indicating protein oxidation. Either intratracheal or intravenous pretreatment of mice with a 9 amino acid peptide called peroxiredoxin 6 inhibitor peptide-2 (PIP-2) significantly reduced all parameters of lung injury by ~50–80%. PIP-2 inhibits NADPH oxidase type 2 (NOX2) activation. We propose that PIP-2 does not affect the mechanically induced lung damage component of VILI but does significantly reduce the secondary inflammatory component.

## 1. Introduction

Adult respiratory distress syndrome (ARDS), now commonly called acute lung injury (ALI), is characterized by diffuse lung injury associated with multiple etiologies that may be either a primary lung affliction or secondary to disease of other organs [1,2,3]. Examples of primary lung diseases that can cause ALI include viral or bacterial pneumonia, aspiration of gastric contents, and inhalation of smoke or other noxious agents while afflictions that may result in secondary ALI include severe trauma especially to the long bones, sepsis, opioid drug overdose, and multiple blood transfusions. Mechanical ventilation of the lungs is routinely used to treat the respiratory failure that frequently accompanies ALI. The parameters of lung ventilation used in the early days of ARDS treatment were designed to obtain close to normal blood oxygenation while maintaining inspired oxygen levels within the non-toxic range [4,5]. This approach can lead to the use of large tidal volumes with resultant high intrapulmonary pressures that can result in lung damage (barotrauma) and augment the lung injury associated with the primary lung disease [6,7,8]. The syndrome of lung damage associated with mechanical over-ventilation has been called ventilator induced lung injury (VILI) [7,9]. The currently accepted standard of care limits barotrauma to the lung by decreasing tidal volume associated with mechanical ventilation to relatively physiological levels [8,9]. This approach to lung mechanical ventilation has had a significant effect on the mortality of ALI, although mortality remains unacceptably high at approximately 30–40% [3,10].

The VILI syndrome has been studied in rodents and other species [11,12,13,14,15]. Most of the studies of VILI have subjected the animals to extended periods (hours) of high volume ventilation while other animal models have utilized a 2-hit approach by administering an inflammatory mediator followed by lung overventilation [11]. However, it is clear that over-ventilation by itself can damage the lung and lead to the ALI syndrome.

The mechanism for tissue injury with VILI has been ascribed to several factors [16]. Most important is the mechanical trauma associated with over-stretching of the lung tissue [17]. This effect can be exaggerated in the presence of ARDS or an underlying diffuse lung disease since the distribution of the inspired gas may be uneven, and some areas of lung tissue may be stretched beyond their elastic limit while other areas are under-ventilated. Further, patients with ALI generally have lung inflammation that can be aggravated by lung overdistention. Thus, mechanical damage to the lung induced by over-inflation in otherwise normal mice can result in a secondary inflammatory response that can potentiate the mechanically induced lung injury [18]. This two-tiered response for induction of VILI with primary lung barotrauma and a secondary injury associated with lung inflammation is analogous to ALI associated with other etiologies [19].

While the lung damage associated with mechanical trauma is easily understood, the precise mechanisms for lung injury associated with inflammation are not as clear [17]. One component is the generation of reactive oxygen species (ROS) that appears to play an important role, both as an initiator of cellular oxidative stress as well as a molecule that signals for over-production of products that may be injurious to the lung [2,17]. NADPH oxidase type 2 (NOX2) is a major source of superoxide anion (a member of the ROS spectrum) in lungs and also in other organs [20,21,22]. The present study is based on the role of NOX2 in generating superoxide and its role in subsequent lung injury.

An important finding in our quest for a pharmacological agent to prevent or treat ALI was that the PLA_2_ activity (called aiPLA_2_) of the enzyme peroxiredoxin 6 (Prdx6) is essential for activation of NOX2 [20,21]. aiPLA_2_ activity generates lysophosphatidylcholine (LPC) that leads through several steps to the mobilization of Rac, an essential step for NOX2 activation [20,21]. We found that a lipid inhibitor (MJ33) of aiPLA_2_ activity or its genetic inactivation were effective in modulating LPS-induced or hyperoxia-mediated ALI in mice, providing evidence that aiPLA_2_ and subsequent NOX2 activation plays an important role in ALI, at least in mice [22,23,24,25,26]. In other studies, we found that lung surfactant protein A (SP-A), a multimeric protein of 248 amino acid monomers, binds to Prdx6 in vitro and inhibits its PLA_2_ activity [26,27,28]. We then described and synthesized the peptide sequence contained in SP-A that was responsible for its Prdx6 binding/NOX2 inhibition [29,30]. This 9 amino acid sequence was called peroxiredoxin 6 inhibitor peptide-2 (PIP-2); it inhibits aiPLA_2_ activity and prevents the activation of NOX2 [30]. Treatment with PIP-2 has resulted in a high level of protection against lung injury in mouse models of ALI or sepsis associated with the administration of bacterial lipopolysaccharide (LPS) [31]. The purpose of the present study was to evaluate possible protection by PIP-2 against lung injury in a mouse model of VILI.

## 2. Materials and Methods

### 2.1. Mice

Male and female C57Bl/6 mice were obtained at approximately 8 weeks of age from Jackson Laboratories (Bar Harbor, ME) and maintained in our vivarium for 1–2 weeks on standard chow. Animals in the vivarium were kept under high-efficiency particulate filtered air (HEPA) maintained at 25 °C ± 3 °C. All procedures involved with the study of mice were approved by the University of Pennsylvania Animal Care and Use Committee (IACUC).

### 2.2. PIP-2

PIP-2 was synthesized by APeptide Corp. (Shanghai, China) as the trifluoroacetate salt at >90% purity as shown by mass spectrometry and HPLC provided by the manufacturer. The amino acid sequence of PIP-2, LHDFRHQIL, has been published previously [30,31]. The peptide was stored at −20 °C, although we have shown stability of the preparation for at least 10 months at room temperature. For administration, the peptide was encapsulated in liposomes [dipalmitoyl phosphatidylcholine (DPPC): egg PC: egg phosphatidylglycerol (PG): cholesterol (0.5:0.25:0.10:0.15, mol fraction] as previously described [31]. The liposomes and PIP-2 were suspended at 14 mg lipid and 2 mg PIP-2 per mL of phosphate-buffered saline (PBS), stored overnight at 4 °C, and used within 24 h of preparation. The acidic calcium-independent phospholipase A_2_ activity (aiPLA_2_) of Prdx6 was measured by radioactivity assay using *sn*-2–^3^H-palmitate DPPC as substrate as described previously [31].

### 2.3. Ventilator Induced Lung Injury (VILI)

To generate a model of VILI, mice were anesthetized with a mixture of ketamine (100 mg/Kg), xylazine (4 mg/Kg), and acepromazine (1 mg/Kg) injected intraperitoneally. To provide airway access for mechanical ventilation, a 20-gauge plastic cannula was inserted into the trachea of the anesthetized mouse, was positioned with the tip at the tracheal carina, and was tied into place with a silk suture. Mice were ventilated with respiratory rate 120/min, positive end-expiratory pressure (PEEP) 2 cm H_2_O, and tidal volume 4 (low) or 12 (high) mL/Kg; the low volume (~0.1 mL) has been our standard tidal volume for mouse studies and the high volume (~0.3 mL) ventilation was expected to cause VILI [32]. When given, PIP-2 in liposomes was administered to mice at the start of mechanical ventilation as a bolus either into the trachea or intravenously (retinal vein) to give 2 µg PIP-2 per gm of body weight, or in one experiment, to give 5 times the dose (10 µg/g body wt.); for control experiments, a similar volume of liposomes that did not contain PIP-2 was given by the same route. The volume of lipid associated with the administration of liposomes was approximately 20–25 µL. After 6 h of mechanical ventilation, the mice were exsanguinated by cutting the abdominal aorta and the lungs were removed from the thorax.

### 2.4. Indices of Lung Injury and Oxidative Stress

After sacrifice of mice, a tissue sample (~0.25 g) was removed from the lung left lower lobe for measurement of wet to dry weight ratio and the lesion was clamped to prevent fluid leakage during lung lavage. Bronchoalveolar lung lavage (BAL) was performed on lungs with 1 mL saline that was slowly infused into the lungs with a syringe and then removed over approximately 1 min. This was repeated twice with fresh saline (3 total lavage procedures /mouse). The BAL fluid (BALf) was used to count the number of nucleated cells (by microscopy) and to measure protein by the Bradford (Coomassie blue) assay. The lung tissue sample was weighed (wet weight) and dry weight was determined after drying the lung sample to constant weight in an oven at 60 °C. Note that these lung samples were not blotted before weighing so that the wet weight and consequently the control wet to dry weight ratio was about 10% greater than the usual values [23,33]. After BAL, the remaining lung was homogenized under N_2_ and stored at −80 °C for subsequent assay of thiobarbituric acid substances (TBARS), 8-isoprostanes, and protein carbonyls as previously described [23,24,25]. The protection efficacy for PIP-2 (in %) was calculated as [1-({VILI+PIP-2}-control)/VILI-control] ×100. In some experiments, cytokine (IL-1β, IL-6, and TNFα) content of BALf was measured by ELISA assay using commercial assay kits (Oxidative Stress Elisa Strip Profiling Assay, Signosis, Santa Clara, CA, USA).

### 2.5. Measurement of ROS in HL60 Cells

HL60 cells, a human blood myeloid cell line that is considered neutrophil-like [34] was purchased from ATCC (Manassas, VA, USA) and grown in RPMI 1640 medium supplemented with fetal bovine serum and non-essential amino acids. Cells were grown in suspension and were used to study the effect of PIP-2 on their ROS production. These cells primarily produce superoxide that is then dismuted to H_2_O_2_. After 16 h in culture, cells were labeled for 30 min. with 5 μM CellROX^TM^ Green, a cell-permeable, non-fluorescent compound under control conditions that is oxidized by reaction with H_2_O_2_ and probably other ROS to a fluorophore that emits green fluorescence upon excitation at 488 nm [35]. Cells were washed to remove excess dye and plated on glass slides, coverslipped, and imaged with a Nikon TMD epifluorescence microscope, equipped with a Hamamatsu L ORCA-100 digital camera, and MetaMorph imaging software (Universal Imaging, West Chester, PA, USA). The fluorescence intensity of images acquired by MetaMorph was integrated within an entire field and background fluorescence was subtracted. Fluorescence intensity was normalized to area and expressed in arbitrary fluorescence units (AFU). These techniques have been reported previously [35,36].

### 2.6. Statistical Analysis

Results are presented as mean ± standard deviation (SD). Group differences for the VILI experiments were evaluated by 2-tailed *t*-test or one-way ANOVA followed by a post hoc *t*-test with Bonferroni correction as appropriate. For studies with HL60 cells, the single-tailed paired *t*-test was used to determine statistical significance. Statistical significance for all studies was accepted as *p* < 0.05.

## 3. Results

### 3.1. Liposome Encapsulation Is Essential for Cellular Uptake of PIP-2

Our previous studies with the isolated and perfused lung showed that liposome encapsulation is an absolute requirement for intracellular uptake of PIP-2 [30]. Thus, our previous studies with the mouse LPS model utilized PIP-2 encapsulated in liposomes for either intratracheal (IT) or intravenous (IV) injection [31]. For the present study, we further evaluated the requirement for PIP 2 encapsulation in liposomes for its intracellular delivery. PIP-2 was encapsulated into our standard liposomes or suspended in PBS immediately prior to IV injection in the anesthetized mouse at a dose of 2 µg/g body wt. After 2 h, mice were sacrificed, and lungs were cleared of blood and homogenized. PIP-2 encapsulated in liposomes inhibited aiPLA_2_ activity of lung homogenate by 84% (*p* < 0.05) while PIP-2 without liposomes (in PBS) had no effect on aiPLA_2_ activity (Table 1). Thus, encapsulation of PIP-2 in liposomes was confirmed as a requirement for its intracellular uptake.

We then evaluated the effect of liposome lipid composition on intracellular delivery of PIP-2. IV injection of PIP-2 encapsulated in our standard liposomes into the anesthetized mouse and studied 2 h later resulted in 81% inhibition of aiPLA_2_ activity in the mouse lung homogenate (Table 2). The substitution of either 75% PC or 75% DPPC for the usual composition (50% DPPC, 25% PC) did not change the effectiveness for intracellular delivery of PIP-2. However, the elimination of PG from the liposome mixture resulted in a decrease of ~12% in effectiveness for PLA_2_ inhibition. Since PG is a negatively charged phospholipid that has been shown previously to facilitate binding and uptake of lipids by lung cells [37], its absence resulting in the loss of the negative charge on the liposomes may be expected to result in decreased liposome uptake.

The liposomes used for the study in Table 2 were prepared with synthesized DPPC while PC and PG were derived from egg. To confirm that the use of egg-derived phospholipids did not influence the results, we compared the effect of liposomes prepared with egg PC (*n* = 2) and egg PG (*n* = 2) to liposomes prepared with synthetic PC and synthetic PG on the ability to deliver PIP-2 intracellularly. The inhibition of aiPLA_2_ activity in the homogenized lung at 2 h after IV injection of liposome-encapsulated PIP-2 for the egg vs. synthesized phospholipid liposomes (*n* = 4) was not statistically different (*p* > 0.05, data not shown).

### 3.2. PIP-2 Administered Intratracheally Abrogates Oxidative Stress Arising from High Tidal Volume-Induced VILI

To study VILI, we examined the effect of low vs. high tidal volume ventilation on the selected indices of lung injury. With low tidal volume ventilation (4 mL/Kg body wt.), the parameters measured were not different from the non-ventilated lungs (non-ventilated control)**,** indicating the absence of lung injury (Table 3). Mice that were ventilated with high tidal volume (12 mL/Kg) for 6 h demonstrated changes consistent with ALI (Table 3). These changes included: (1) an increase in nucleated cells in lung lavage fluid compatible with lung inflammation; (2) an increase in protein content in lung lavage fluid and increased lung wet to dry weight ratio indicating increased alveolar permeability and lung edema; and (3) increased lung TBARS, 8-isoprostanes, and protein carbonyls indicating oxidative stress with the oxidation of both lipid and protein components of lung tissue. Since lungs from mice ventilated under low (usual) volume conditions showed no difference from control (normal lungs from spontaneously breathing mice) in these tissue parameters, hyperventilation was the cause of the lung injury.

Thus, the hyperventilated mice serve as a model for VILI. The treatment of mice with intratracheal PIP-2 (2 µg/g body wt.) at the start of ventilation significantly inhibited the lung injury (VILI) associated with lung overventilation. A 5-fold increase in the amount of PIP-2 delivered at the start of ventilation (10 µg/g body wt.) had no additional effect over the 2 µg/g dose (Table 3). The degree of “protection” against the various parameters of lung injury ranged from ~50% for BALf PMNs to ~70–80% for BALf protein and for markers of oxidative stress.

### 3.3. Intravenous vs. Intratracheal Administration of PIP-2

We compared the efficiency of IV vs. IT administration of PIP-2. The degree of protection with IV administration of PIP-2 (Table 4) was similar to that shown with IT administration of the agent (Table 3). Subsequent studies utilized IV PIP-2.

An increased release of various cytokines has been associated with ALI [38]. We selected 3 cytokines for analysis in the BAL fluid from the mouse model of VILI. The greatest increase of the 3 cytokines that were measured was seen with IL-6 while a lesser increase was seen with TNF-alpha; values for these two cytokines increased significantly with VILI while values not significantly different from control were seen with PIP-2 treatment (Figure 1). The lavage value for IL-1β did not change significantly with VILI or PIP-2 treatment.

### 3.4. Male vs. Female Mice

The results shown in Table 3; Table 4 were obtained using male mice. Subsequently, female mice also were studied using the same protocol as for male mice with IV injection of PIP-2. Female mice exhibited indices of VILI and protection by PIP-2 (Table 5) that were not significantly different from those shown by male mice (Table 4).

### 3.5. Effect of PIP-2 on ROS Production in Human Cells

We have demonstrated previously that the mechanism for the effectiveness of PIP-2 in the amelioration of lung injury reflects the inhibition of NOX2-mediated ROS production [23,25,31]. These studies were carried out with mouse lungs. In order to study whether PIP-2 affects the same pathway in humans, we studied NOX2 activation in HL60 cells, a human neutrophil-like cell line. Treatment of the cells with PIP-2 blocked ROS production in response to treatment with LPS (Figure 2). Thus, PIP-2 effectively inhibits ROS production by human PMN-like cells similar to its effect in mouse lungs.

## 4. Discussion

The use of high tidal volume ventilation of mice represents a model for acute lung injury due to mechanical trauma. The accepted mechanism for the syndrome is that lung damage reflects mechanical stress associated with over stretching of lung connective tissue as well as other lung tissue components; this leads to disruption of the alveolar–epithelial and the endothelial–capillary barriers leading to fluid accumulation in the lung [17,39]. While the mechanically induced lung damage can be of varying severity, it can result in a secondary inflammatory response that amplifies the tissue injury; this secondary response results in inappropriate activation of leukocytes and platelets and enhanced activation of coagulation signals [17,39]. This mechanism is similar to the proposed mechanism for other etiologies of ALI, that is, a primary insult followed by a secondary inflammatory response [3,11]. For example, ALI can result from the aspiration of gastric contents resulting in a chemical injury (burn) followed by an inflammatory response while VILI results from mechanical injury followed by an inflammatory response. Interestingly, when mechanical ventilation is used to treat ARDS, the lung has already been injured by a secondary inflammatory component resulting from the inciting disease, so that the inflammatory component of VILI would represent an addition to the process that led to the institution of mechanical ventilation. Although drug treatment (e.g., with PIP-2) would not be expected to affect the lung injury associated with mechanical (or chemical) trauma, it might alter the secondary inflammatory component as shown in the present study. The bottom line is that whatever the etiology for development of ARDS, treatment directed to the primary initiating process is necessary, although it may not be effective, while PIP-2 can combat the potentially more serious secondary inflammatory component.

We have proposed that the secondary inflammatory response mediating ALI reflects, in large part, an important role for ROS. Release of ROS has a two-fold effect, namely: (1) direct tissue damage due to oxidation of tissue lipid and protein components, and (2) cell signaling for the release of cytokines and other inflammatory mediators. There are many sources for ROS in the lung, but a major source is through activity of the enzyme NADPH oxidase (NOX). There are 7 enzymes in the NOX family that can produce either superoxide anion or H_2_O_2_ depending on the NOX type [40]. NOX2, a sub-type that produces superoxide anion, is widely distributed in tissues and is responsible for much of the ROS production in lung [40,41,42]. Thus, in our previous studies in mice of both LPS-mediated and hyperoxia-mediated lung injury, prevention of NOX2 activation largely prevented lung tissue injury [23,24,25,31]. In these previous studies, NOX2 activation was prevented by treatment with an inhibitor of aiPLA_2_ activity, either MJ33, a lipid inhibitor, or PIP-2, a peptide inhibitor [23,31]. Both MJ33 and PIP-2 exert their inhibitory effect by binding to Prdx6, thereby inhibiting the aiPLA_2_ activity that is required for NOX2 activation. LPS-mediated injury in the mouse model also was prevented by genetic inactivation of aiPLA_2_ activity [25]. Since aiPLA_2_ inhibition would not be expected to significantly alter the lung injury associated with the direct effects of mechanical trauma, we assume that the treatable inflammatory-mediated injury was responsible for >50% of lung injury in this model of VILI.

As described above, our studies related to PIP-2-mediated therapy for ALI began with the finding that the lung surfactant protein SP-A binds to Prdx6 [27] and inhibits its aiPLA_2_ activity [28,29]. SP-A is a protein of ~26 kD monomeric molecular weight comprising 248 amino acids [43]. PIP-2 is a 9 amino acid peptide that represents the molecular sequence in human SP-A for its binding to Prdx6 and comprises only 1.3% of the monomeric SP-A amino acids [31]. Physiologically, Prdx6, mainly in its phosphorylated state and through its phospholipase A_2_ activity, generates lysoPC that in turn activates Rac1, a crucial component of the NOX2 activation complex [21]. Thus, in the presence of PIP-2, cell membrane-bound NOX2 cannot be activated. Since NOX2 is the major source of ROS in many lung cells, the ROS “load” associated with ALI should be considerably decreased in the presence of PIP-2, based on the studies in mice. Evidence that the pathway for PIP-2 inhibition of NOX2 activation is similar in mice and humans is shown with the PIP-2-mediated inhibition of ROS production by a human polymorphonuclear leukocyte-like cell line in the present study. Previous studies also have shown that Prdx6 is required for in vitro activation of NOX2 in a variety of human cell lines [44,45,46].

While the inhibition of NOX2 activation may represent a major effect of PIP-2 treatment, other pathways also may be involved in PIP-2 mediated protection against lung injury, although none of these have as yet been confirmed. For example, PIP-2 blocks generation of active Rac1 that is the basis for inhibition of ROS generation, as described above. However, Rac1 is involved in many physiological processes including the regulation of proteins that mediate the stretch-induced increase in alveolar–capillary permeability [47]; increased alveolar-capillary permeability will increase lung fluid and contribute to lung edema. Thus, inhibition of Rac1 activation by PIP-2 may protect against lung injury by preventing the Rac1 mediated increase in alveolar–capillary permeability of the lung. However, the significance of this pathway in VILI and its relationship to the NOX2 pathway remain to be investigated.

What are the practical aspects of the present results as related to mechanical ventilation of human patients in the intensive care unit? It is generally accepted that for most patients, mechanical ventilation should utilize ventilatory parameters that reflect close to normal ventilation in order to prevent VILI. This approach has been called “lung protective ventilation” (LPV) [48]. However, LPV may not resolve the hypoxemia situation in some patients and the dilemma is whether to accept the potentially deleterious effects of high-volume ventilation (VILI) or to continue LPV using a higher concentration of inspired oxygen that could lead to oxygen toxicity [49,50,51]. Of note, ischemic areas of lung associated with altered blood flow as occurs with lung injury may be especially susceptible to lipid peroxidation associated with elevated inspired O_2_ concentration [52]. Similarly, while a mild degree of hypercapnia may be tolerated, ALI patients with more severe hypercapnia (PaCO_2_ ≥ 50 mmHg) had higher complication rates, more organ failures, and worse outcomes [53]. To address these situations, additional treatment methods such as extra-corporeal membrane oxygenation (ECMO) have been proposed [48]. The present study suggests that if the usual LPV parameters are inadequate to maintain approximately normal blood gas concentrations, the use of PIP-2 concurrently with cautious initiation of more aggressive mechanical ventilation may prevent the inflammatory component associated with over-stretching of the lung. This recommendation assumes that the secondary inflammatory component may be responsible for the major share of lung injury.

While the use of PIP-2 on the initiation of high volume mechanical ventilation is supported by this study, what about its use after lung damage has already occurred? Our previous studies with LPS and hyperoxia as the agonists in mice showed that PIP-2 treatment allowed the lungs that had been damaged as a result of inflammation to heal almost completely within several hours [23,31]. Thus, initiation of PIP-2 treatment in the presence of inflammatory lung injury can be predicted to allow reversal of the injury, but that speculation was not tested in the present experiments.

PIP-2 has been proposed as a therapy for lung inflammation associated with ARDS [31] and its efficacy in VILI likely reflects the role of inflammation in that syndrome. Unfortunately, there have been no other therapeutic options that have proven uniformly effective or have been approved for the treatment of ARDS in humans, whether due to VILI or other etiologies. Therapeutic modalities that have been proposed for treatment of VILI based on animal studies include the cholesterol-lowering agent simvastatin [32], the inhaled anesthetic sevoflurane [54], bixin to target NRF2 [55], neuromuscular blocking agents to reduce ventilator-patient asynchrony [56], H_2_S to regulate ER stress [57], microRNA 146a for altered mechanotransduction [58], dexmedetomidine to activate ERK 1,2 [59], alpha1-antitrypsin to inhibit neutrophil elastase [19], and various anti-inflammatory agents including ginsenoside [12] and lipoxin A4 [60], but none of these agents or procedures has been approved for use in the ICU. Based on the mouse studies of acute lung injury, PIP-2 represents a potential therapeutic agent for treating or preventing VILI. Importantly, PIP-2 is a naturally occurring product derived from the lung SP-A protein that is predicted to be non-antigenic [30] and is expected to be non-toxic, although that, of course, will require further study in patients. Assuming non-toxicity of the agent, PIP-2 could become an essential component of the mechanical ventilation regimen by preventing or treating the inflammatory component of lung injury associated with high tidal volume ventilation.

## 5. Conclusions

A model for lung injury associated with over-ventilation (VILI) was developed using high tidal volume mechanical ventilation of anesthetized mice. Injury in male and female mice was similar. The degree of lung injury was decreased 50–80% by either intravenous or intratracheal infusion of the nonapeptide PIP-2, an inhibitor of the activation of lung type 2 NADPH oxidase. We have interpreted these results by postulating a 2-hit model of mechanically induced lung injury followed by a secondary injury due to lung inflammation. We propose that PIP-2 blocks the inflammatory component while having relatively little effect on the mechanical disruption of the lung. This agent (PIP-2) may be considered for the prevention of the inflammatory manifestations of VILI in ventilated patients when the standard lung-protective ventilation protocol is not sufficient to maintain adequate blood oxygenation.

## Figures and Tables

**Figure 1 antioxidants-10-00925-f001:**
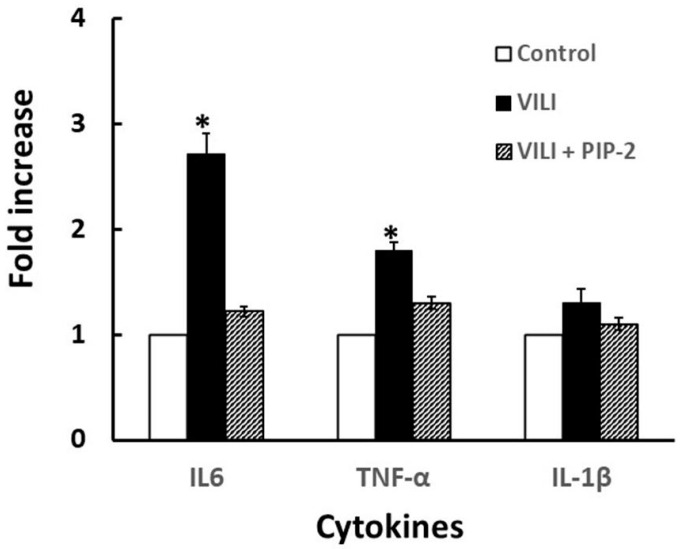
Effect of PIP-2 on cytokines release into BALf. Male mice were studied under conditions described in Table 4. Mice were unventilated (control) or ventilated with high tidal volume (VILI) ± PIP-2 for 6 h. After sacrifice, lungs were lavaged with normal saline and the BAL fluid was analyzed for cytokines. Data is presented as mean ± SD for *n* = 4 mice, * *p* < 0.05 vs. corresponding control or VILI values.

**Figure 2 antioxidants-10-00925-f002:**
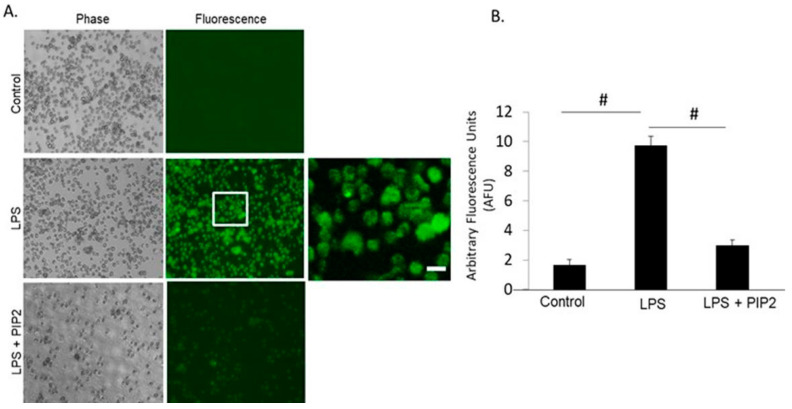
Inhibition of H_2_O_2_ production by PIP-2 in HL60 cells. (**A**). Fluorescent images of HL60 cells under control (liposomes alone), LPS + liposomes, or LPS + PIP-2 in liposomes. Images were acquired at λexcitation = 488 nm. A. All images were acquired with the same exposure and acquisition settings. The area enclosed by the box in the middle panel is shown at higher magnification in the far-right panel. Scale bar is 10 µm. (**B**). The integrated fluorescence intensity generated by HL60 cells across several microscopic fields (*n* = 3) was quantified using MetaMorph Imaging software. Intensity of fluorescence was normalized to area. The data is expressed as mean ± SD for *n* = 3 independent experiments. # *p* < 0.001 for the indicated comparisons.

**Table 1 antioxidants-10-00925-t001:** aiPLA_2_ activity of mouse lung after injection of PIP-2. Mice were anesthetized and injected intravenously with PIP-2 at 2 µg/g body wt. Lungs were analyzed 2 h later for aiPLA_2_ activity. Results are mean ± SD for *n* = 3. * *p* < 0.05 vs. Control and vs. PIP-2 mixed with phosphate buffered saline (no liposomes).

	aiPLA_2_ Activity nmol/hr/mg Protein
Control (Liposomes only)	8.81 ± 0.42
PIP-2 (Encapsulated in Liposomes)	1.55 ± 0.03 *
PIP-2 (No Liposomes)	8.43 ± 0.40

**Table 2 antioxidants-10-00925-t002:** Effect of liposomes composition on delivery of PIP-2 to lungs. PIP-2 in liposomes was delivered to mice by a bolus IV infusion. After 2 h lungs were cleared of blood and PLA_2_ activity of the lung homogenate was measured. % inhibition = value relative to no PIP-2. * % of total lipid; all liposomes also contained 15% cholesterol. Ϯ Mean ± SD for *n* = 3 or ± range for *n* = 2.

Composition * DPPC: PC: PG	PIP-2	*n*	Activity ϯ nmol/mg prot/hr	Inhibition %
50/25/10	NO	2	8.83 ± 0.10	-----
50/25/10	YES	3	1.64 ± 0.07	81.4
0/75/10	YES	3	1.72 ± 0.10	80.5
75/0/10	YES	2	1.84 ± 0.02	79.2
55/30/10	YES	2	2.51 ± 0.12	71.7

**Table 3 antioxidants-10-00925-t003:** Effect of intratracheal (IT) PIP-2 on manifestations of VILI. Male mice were ventilated for 6hr with tidal volume of 4 (low) or 12 (high, VILI) mL/Kg, respiratory rate 120/min, PEEP 2 cm H_2_O. Mice were administered PIP-2 (2 or 10 µg/g body weight) encapsulated in liposomes by IT bolus infusion. After 6 h, mice were sacrificed, and lungs processed as described in Materials and Methods. Results are mean ± SD for *n* = 4. * *p* < 0.05 vs. all other groups. ϯ *p* < 0.05 vs. Control, Low tidal volume (±PIP-2) and VILI. There are no significant differences between 2 and 10 ug/g PIP-2.

Condition	# of cells in BALf (×10^4^/g body wt)	Total Protein in BALf (mg/g Body wt	Wet/Dry Weight Ratio of Lung	TBARS pmol/mg prot.	8—Isoprotanes pmol/mg prot	Protein Carbonyls nmol/mg prot
Control (non-ventilated)	0.97 ± 0.12	76.9 ± 4.2	5.61 ± 0.04	74.6 ± 4.6	33.1 ± 6.0	5.63 ± 0.48
Low tidal volume	1.03 ± 0.06	80.1 ± 4.4	6.02 ± 0.26	81.7 ± 3.0	37.8 ± 4.2	5.90 ± 0.46
Low tidal volume + PIP-2	1.01 ± 0.08	78.8 ± 3.6	5.98 ± 0.22	79.9 ± 4.2	36.2 ± 2.4	5.82 ± 0.48
High tidal volume (VILI) *	19.6 ± 2.84	235 ± 20	10.8 ± 0.42	530 ± 14.4	174 ± 9.8	22.3 ± 1.56
VILI + PIP-2 (2 µg/g) ϯ	8.4 ± 0.62	102 ± 16.2	8.14 ± 0.22	206 ± 24	75.9 ± 2.0	8.85 ± 0.84
VILI + PIP-2 (10 µg/g) ϯ	8.1 ± 0.46	107 ± 16.2	7.62 ± 0.36	202 ± 12.4	76.5 ± 2.4	8.12 ± 0.34

**Table 4 antioxidants-10-00925-t004:** Effect of intravenous (IV) PIP-2 on manifestations of VILI. Male mice were ventilated for 6 h with tidal volume of 12 mL/Kg, respiratory rate 120/min, PEEP 2 cm H_2_O. Control mice breathed normally (no mechanical ventilation). Mice were administered PIP-2 (2 µg/g body weight) encapsulated in liposomes by IV bolus injection. Mice were sacrificed after 6h, and lungs were cleared of blood, lavaged, and processed as described in Materials and Methods. Results are mean ± SD for *n* = 4. * *p* < 0.05 vs. Control and vs. VILI+ PIP-2; ϯ *p* < 0.05 vs. Control. Statistics represent all values in the indicated row.

Condition	# of Cells in BALf (×10^4^/g body wt)	Total Protein in BALf (mg/g Body wt	Wet/Dry Weight Ratio of Lung	TBARS pmol/mg prot.	8—Isoprostanes pmol/mg prot	Protein Carbonyls nmol/mg prot
Control (non-ventilated)	1.04 ± 0.14	82.4 ± 10.0	5.94 ± 0.50	80.5 ± 3.2	31.5 ± 5.4	5.48 ± 0.58
VILI *	18.9 ± 1.16	218 ± 19	10.4 ± 0.70	512 ± 26	178 ± 34	21.1 ± 1.32
VILI + PIP-2 ϯ	8.17 ± 0.28	103 ± 5.6	7.4 ± 0.24	186 ± 36	81.3 ± 7.2	8.44 ± 1.38

**Table 5 antioxidants-10-00925-t005:** Effect of PIP-2 on manifestations of VILI in female mice. Female mice were ventilated with parameters the same as for male mice (tidal volume 12 mL/Kg, respiratory rate 120/min, and PEEP 2 cm H_2_O). Mice were administered PIP-2 (2 µg/g body weight) encapsulated in liposomes by an IV bolus injection. Mice were sacrificed after 6 h, and lungs were cleared of blood, lavaged, and processed as described in Materials and Methods. Results are mean ± SD for *n* = 4. * *p* < 0.05 vs. Control and vs. VILI + PIP-2; ϯ *p* < 0.05 vs. Control and vs. VILI. Statistics represent all values in the indicated row.

Condition	# of Cells in BALf (×10^4^/g Body wt)	Total Protein in BALf (mg/g Body wt	Wet/Dry Weight Ratio of Lung	TBARS pmol/mg prot.	8—Isoprostanes pmol/mg prot	Protein Carbonyls nmol/mg prot
Control (non-ventilated)	1.01 ± 0.06	81.7 ± 10.6	5.72 ± 0.24	81.7 ± 2.0	39.6 ± 1.82	5.83 ± 0.24
VILI (no PIP-2) *	17.7± 0.46	222 ± 16	9.85 ± 0.54	479 ± 28	183 ± 22	20.6 ± 2.8
VILI + PIP-2 ϯ	7.67 ± 0.58	93.6 ± 7.0	7.49 ± 0.34	180 ± 36	85.2 ± 5.6	8.87 ± 0.20

## Data Availability

The data presented in this study are available on request from the corresponding author.

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
