# Peer review of "A Peptide Inhibitor of Peroxiredoxin 6 Phospholipase A2 Activity Significantly Protects against Lung Injury in a Mouse Model of Ventilator Induced Lung Injury (VILI)"

_antioxidants, 2021, doi:10.3390/antiox10060925_

Round 1

Reviewer 1 Report

This work is addressing a key problem concerning the treatment of lung lesion  requiring mechanical ventilation. May I suggest to extend in the introduction the sequential phases leading to the development of a severly edematous lung.  This would provide a more extended validation of the study to other cases ending in severe respiratory distress. The authors might quote some papers from the Milano group concerning the damage (fragmentation) of the lung interstitial matrix leading to severe lung edema  (just  a couple of author names I came across in reviewing this paper but feel free to browse on your own : Miserocchi G, Negrini D). I believe it is important to let clinicians know that preventive steps ought to be considered before severe respiratory distress explodes. Most of clinicians simply ignore this step. The authors report data on W/D ratio and proteins in the BAL.  Control non ventilated  exhibits a W/D in the higher range (5.6 compared to a reported control value of 4.95-5). However data  clearly indicate the benefit of Low VT with PIP-2. A point that could be made in the discussion  is a comparison with the W/D ratios reported in the literature as results of several ventilatory strategies considered as successfull: they never go below 7! May I  comment that the W/D ratio of VILI+ PIP-2 of 8.14 with proteins in the BAL is still a very bad lung indeed. Fig.1 and 2  are very convincing. The discussion line 323-339 is very well written. 

Reviewer 2 Report

The authors Fisher et al. describe the effect of a new peptide inhibitor to reduce lung injury in a mouse model of ventilated induced lung injury (VILI) and its effect on the ROS generation inhuman HL60 cells.

Although authors could show an effect on the induction of lung injury after VILI there is some points to improve:

  • Mayor:
  • The connection to oxidative stress is not clear. Therefore some improvement would be necessary:
  • To detect oxidative stress in the lung authors could perform western blot for 4 hydroxynonenal which is an aldehydic product of lipid peroxidation and a marker for oxidative stress in the tissue. In addition and to see the spatial areas of oxidative stress, authors could stain the lung tissues with an antibody against 8-oxo-7,8-dihydroguanosine (8-OHG) which is an oxidized guanosine found in DNA and RNA damaged by oxidative stress and another marker for cellular oxidative stress.
  • As neutrophils produce primary superoxide instead of H2O2. To distinguish between H2O2 and superoxide generation, authors should repeat the measurement with H2O2 specific methods such as AmplexRed. Extracellular superoxide generation as it occurs in neutrophils can be easily detected by Cytochrome C reduction assays. In both cases specific inhibitors such as Catalase (H2O2 scavenger) and Superoxide Dismutase (superoxide scavenger) should be added.
  • Oxidative stress in the lung can not only derive from neutrophils, but from other lung cells. ROS measurement could be performed with other cell lines e.g. A549 which is a lung epithelial derived cell line.

  • Minor:

  • The term PIP-2 used for the peroxiredoxin 6 inhibitor peptide-2 is a common abbreviation for the phospholipide phosphatidylinositol-4,5-bisphosphate and therefore a change of the name should be considered.
  • The description of the statistic should be improved in the figures and tables by adding in all tables and figures the sample size, the type of error shown and the p-value.
  • Authors wrote they measured H2O2, but for detection CellROX green was used which is not H2O2 specific. Please change your text accordingly.

Reviewer 3 Report

This study tests if PIP-2, a peptide mimicking SP-A region responsible for inhibition of Prdx6 and subsequent PLA2/NOX2 activation, protects against ventilator induced lung injury (VILI) in an adult mouse model. PIP-2 liposomal packaging was first optimized based on inhibition of pulmonary PLA2 activity and then intratracheal or intravenous PIP-2 delivery occurred prior to 6 h VILI with numerous lung injury/oxidative stress outcomes. The data demonstrate that PIP-2 attenuates injury parameters at this single timepoint but there are remaining questions regarding spatial distribution of lung injury (not complete protection) and temporal relevance during pathogenesis with several examples of over-reaching statements surrounding molecular mechanisms. The data have strong interest and addressing these concerns will strengthen the study.

Major concerns

  1. Data are presented as mean±SEM for animal studies which is not correct. SEM should be used to represent the spread between means from different sets of experimental replicates and for these experiments each adult mouse represents an experimental replicate. Therefore, mean±SD is the most appropriate way to represent the data.
  2. Although one-way ANOVA with Bonferroni posthoc analysis is a good way to analyze the set of data, significance is only indicated between VILI and VILI+PIP-2 groups without any comparisons between VILI+PIP-2 and control. These statistical comparisons should be included because much of the manuscript (including the title) is very liberal with stating PIP-2 protection against VILI when it appears that PIP-2 provides partial protection which may be restricted to specific injury parameters. If so, that is very interesting to speculate about possible SP-A/NOX2-independent modes of VILI-induced injury – perhaps this is due to other sources of hydrogen peroxide and other reduced/reactive oxygen species.
  3. Why was 6 h selected as the timepoint for analysis following VILI? There were no time course studies performed so it’s difficult to understand what this time point represents in terms of pathogenic development. But it is increasingly difficult to understand if PIP-2 prevents or delays injury parameters associated with VILI. In other words, are the protective effects of PIP-2 short-lived? Furthermore, the experimental design could have been improved by performed a dose-response study with decreased concentrations of PIP-2. Unclear why 10 ug/g was selected when 2 ug/g inhibited 84% aiPLA2 activity. Lastly, what are the weaknesses of using a control lacking peptide delivery (could have used a scrambled peptide instead)?
  4. Although the lung injury parameters analyzed support the work, they are by no means complete and the study should provide pathological analysis. For example, since it doesn’t appear that PIP-2 completely protects against VILI, this raises questions regarding protection. Does PIP-2 protect specific regions of the lung against VILI or is there just global attenuation of lung injuries? Pathology is essential.
  5. Wet/dry ratio was performed on a subjective portion of lower left lung – was this piece of lung taken before or after the aorta was severed? This could cause artifacts in the data.
  6. Is CellROX green truly specific for hydrogen peroxide (Figure 2 legend title)? I don’t believe this has been conclusively shown and Invitrogen markets CellROX green as a general ROS detecting fluorophore. Saying that hydrogen peroxide is being detected is very misleading.
  7. The discussion needs to also highlight limitations of the study. Many of the conclusions are a gross oversimplification of the results without any awareness of weaknesses/caveats based on the experimental design and model system. Areas to include in this discussion are mentioned throughout this review.
  8. There are numerous areas in the manuscript which assume that PIP-2 works through a SP-A/Prdx6/NOX2 axis and no such studies supporting this conclusion are presented in the manuscript (knockout mice, other inhibitors, etc.). First the language needs to be accurate and if these studies were done previously with PIP-2 they must be cited with an assessment of the rigor.

Minor concerns

  1. Clarify if the mouse strain is C57Bl/6 or C57Bl/6J. These strains have very different responses to lung injury models because of the NNT mutation in C57BL/6J.
  2. Be consistent with using correct tidal volume units (mL/kg). See abstract for an example.
  3. Good that an experiment was repeated with female mice to consider sex as a variable. However, line 252 states that data generated from female mice were ‘not significantly different’ from male mice. Is this just an observation or were statistical comparisons performed?
  4. The first paragraph of the Introduction contains largely irrelevant information and should be re-written.
  5. Take caution when describing PIP-2 dosing because pulmonary drug concentration is different than how much was dosed in the animal.

Reviewer 4 Report

General comments:

The authors conclude that inhibition of the production of superoxide anions by NOX2 significantly reduced the lung injury associated with high volume ventilation (VILI) by NOX2 inhibitor (PIP-2) in animal model. However, no tissue image was provided to show the recovery for lung injuy by using PIP-2.

Major comments:

  1. The authors demonstrate that Inhibition of H2O2 production by PIP-2 in HL60 cells based on Fluorescent images at Figure 2. It will further provide a supporting evidence if the flow cytometry is performed.
  2. The authors claimed that PIP-2 acts via blocking NOX2 activity based on the finding for ROS intensity change (Figure 2). However, this evidence may by an indirect data. The western blotting for NOX2 expression may provide a direct evidence for claiming NOX2-blocking activity of PIP-2.
  3. Although the authors claimed that “A Peptide Inhibitor of Peroxiredoxin 6 Phospholipase A2 Activity Protects Against Lung Injury in a Mouse Model”, no tissue image was provided to show the recovery for lung injury by using PIP-2.

Minor comments:

  1. Although the authors cite the reference that the amino acid sequence of PIP-2 has been published previously [23], it will be better to list this 9-amino acid sequences again.

Round 2

Reviewer 2 Report

The authors show effort to answer my questions and do not have any further comments.

Reviewer 4 Report

All reviewer's concerns have been well responded to.